# Effects of Biochar Applied in Either Rice or Wheat Seasons on the Production and Quality of Wheat and Nutrient Status in Paddy Profiles

**DOI:** 10.3390/plants12244131

**Published:** 2023-12-11

**Authors:** Zirui Chen, Jiale Liu, Haijun Sun, Jincheng Xing, Zhenhua Zhang, Jiang Jiang

**Affiliations:** 1Co–Innovation Center for Sustainable Forestry in Southern China, Nanjing Forestry University, Nanjing 210037, China; chenzr@njfu.edu.cn (Z.C.); liujiale@njfu.edu.cn (J.L.); jiangjiang@njfu.edu.cn (J.J.); 2Institute of Jiangsu Coastal Agricultural Sciences, Yancheng 224002, China; sdauxxx@163.com; 3School of Agriculture and Environment, The University of Western Australia, Crawley, WA 6009, Australia

**Keywords:** amino acid, biochar, crop rotation, soil fertility, wheat yield

## Abstract

In a rice–wheat rotation system, biochar (BC) applied in different crop seasons undergoes contrast property changes in the soil. However, it is unclear how aged BC affects the production and quality of wheat and the nutrent status in a soil profile. In the present soil column experiment, the effects of no nitrogen (N) fertilizer and BC addition (control), N fertilizer (N420) and BC (5 t ha^−1^) applied at rice [N420 + BC(R)], or wheat [N420 + BC(W)] seasons at a same rate of N fertilizer (420 kg ha^−1^ yr^−1^) on yield and quality of wheat as well as the nutrient contents of soil profiles (0–5, 5–10, 10–20, 20–30, 30–40, and 40–50 cm) were observed. The results showed that N420 + BC(W) significantly reduced NH_4_^+^-N content in 5–10 and 10–20 cm soils by 62.1% and 36.2%, respectively, compared with N420. In addition, N420 + BC(W) significantly reduced NO_3_^−^-N contents by 17.8% and 40.4% in 0–5 and 20–30 cm profiles, respectively, but N420 + BC(R) slightly increased them. The BC applied in wheat season significantly increased the 0–5 and 40–50 cm soil total N contents (24.0% and 48.1%), and enhanced the 30–40 and 40–50 cm soil-available phosphorus contents (48.2 and 35.75%) as well as improved the 10–20 and 20–30 cm soil-available potassium content (38.1% and 57.5%). Overall, our results suggest that N420 + BC(W) had stronger improving effects on soil fertility than N420 + BC(R). Compared to N420, there was a significant 5.9% increase in wheat grain yield, but no change in total amino acids in wheat kernels in N420 + BC(W). Considering the responses of soil profile nutrient contents as well as wheat yield and quality to BC application in different crop seasons, it is more appropriate to apply BC in wheat season. Our results could provide a scientific basis for the ideal time to amend BC into the rice–wheat rotation system, in order to achieve more benefits of BC on crop production and soil fertility.

## 1. Introduction

Nitrogen (N) is one of the essential plant nutrients and an important agricultural output-limiting factor [1]. Applying N fertilizer is therefore a crucial practive in raising crop yield and quality. However, excessive use of N fertilizer not only lessens or even eliminates the effect of N in increasing crop yield and economic benefits, but also causes serious N losses, which threaten the atmospheric and aquatic environment [2,3]. To avoid this, N use efficiency (NUE) can be improved by applying biochar (BC) without increasing the amount of inorganic N fertilizer or even under reducing conditions to ensure crop yields and reduce N losses. This has been widely recognized as an approach to reduce production costs, improve crop yield, and protect the environment [4,5,6]. According to previous reports, BC is a stable and difficult-to-degrade organic carbon compound which is produced by pyrolysis under low oxygen and high-temperature environments and can be used as a soil conditioner [7,8] to increase and conserve soil fertility [9] as well as improve crop NUE and yield [10,11].

Rice–wheat rotation is the basis for the year-round farming pattern of the Yangtze River Delta, a major grain-producing region in China [12]. To ensure crop yields and reduce production costs, soil tillage can be performed at the initiation of rice or wheat season under a rice–wheat rotation production system. In farming practice, BC is always amended in conjunction with soil tillage to reduce production expenses. This means that BC could be applied in either rice or wheat season [13,14,15]. Although BC is chemically very stable, it is still subject to aging changes through chemical reactions, water erosion, and microbial decomposition, especially when the residence time of BC in the soil is prolonged [16,17]. The properties of BC, such as pH, surface elemental composition, and non-aromatic structure, are significantly changed along with the aging process [18,19]. Apparently, the moisture status of paddy soil varies significantly during the growth cycle of rice and wheat, and the unique alternating wet and dry (aerobic–anaerobic) water management leads to the consequent different changes in the nature and function of the BC added to the paddy soil [20,21]. However, there is a lack of study on whether BC applied at different crop seasons has differential effects on soil nutrients and crop yields.

Previous studies have demonstrated that BC addition impacted the soil nutrients in the tillage layer (0–20 cm), either by increasing or sequestering them. For example, Jing et al. [22] found that the soil organic matter (SOM) in paddy fields increased remarkably following the BC addition. Han et al. [23] found that the SOM, NO_3_^−^-N, and total N increased significantly in 0–20 cm soil after BC addition. In addition, it has also been found using soil profile depth that BC particles migrate from topsoil to subsoil [24], which in turn affects subsoil properties. Ding et al. [25] showed that 39–51% of BC would remain in the top 30 cm of soil after 11 years, and significantly increased the soil organic carbon (SOC) content in 0–30 cm soil. In particular, periodic drying/re-wetting conditions under rice–wheat rotation systems can lead to the presence of cracks and pores in the soil, making downward movement of BC more pronounced and subsequently altering subsoil properties [26]. Nevertheless, most previous studies have only explored the effects of BC application in the same season on the soil or crop, and there is limited study on the effects of BC application in different seasons on soil profile nutrient content and crop yield. We hypothesized that the application of BC in rice–wheat rotation system has different effects on fertility in the soil profile and can significantly affect the yield and quality of wheat. Therefore, this study intended to investigate the effects of BC application in rice or wheat season on crop yield, NUE, grain quality, and post-harvest soil profile fertility characteristics in wheat season in a rice–wheat rotation system. The results of this study will provide a theoretical basis for appropriate BC application.

## 2. Results

### 2.1. Soil Properties with Profile Depth

#### 2.1.1. pH

Overall, aside from N420, soil pH showed a decreasing trend with profile depth. Compared with the control treatment, the two added BC treatments showed significant decreases in soil pH across all profiles, such as pH decreasing from 7.07 to 6.80 in the N420 + BC(R) treatment, and pH decreasing from 7.04 to 6.87 in the N420 + BC(W) treatment (Figure 1). The pH was not significantly altered in the N420 + BC(R) and N420 + BC(W) treatments compared to the N420 treatments at 0–5, 5–10, and 10–20 cm profile depths, but the pH was significantly lower in the N420 + BC(R) and N420 + BC(W) treatments than the N420 treatment at 20–30, 30–40, and 40–50 cm profile depths. Compared to N420, the pH in N420 + BC(R) and N420 + BC(W) treatments decreased by 0.26 and 0.19 units, respectively, at 40–50 cm depth. The magnitude of pH across the profile was control > N420 > N420 + BC(W) > N420 + BC(R). Generally, the pH in N420 + BC(R) and N420 + BC(W) treatments showed a significant reduction in the subsoil profile (20–50 cm).

#### 2.1.2. NH_4_^+^-N, NO_3_^−^-N, and Total N Contents

Soil NH_4_^+^-N contents tended to increase from the 0–5 cm to 5–10 cm profile and thereafter decreased with increasing depth of the soil profile (Figure 2a). For the top 5–10 cm soil, the NH_4_^+^-N contents were in orders of control > N420 > N420 + BC(R) > N420 + BC(W). Compared to N420, the NH_4_^+^-N contents were reduced in soil profiles of the BC-amended treatments, with 43.9–62.1%, 33.8–91.5%, and 66.8–93.1% significantly lower NH_4_^+^-N contents in the 5–10, 20–30, and 40–50 cm soil profiles, respectively. Moreover, NH_4_^+^-N contents were significantly lower at the 5–10, 10–20, and 30–40 cm profiles in N420 + BC(W) than in N420 + BC(R).

Soil NO_3_^−^-N contents showed a trend of first decreasing, then increasing, and again decreasing trend along with the profile depth. In general, soil NO_3_^−^-N contents were significantly lower in fertilizer N-applied treatment than the control (Figure 2b). For instance, soil NO_3_^−^-N contents in three N-applied treatment were 78.3%, 84.2%, and 65.2% significantly lower, respectively, than in the control treatment at the 5–10 cm profile. Compared to N420, N420 + BC(W) significantly reduced NO_3_^−^-N contents by 17.8% and 40.4% at the 0–5 and 20–30 cm soil profiles, respectively, but increased NO_3_^−^-N contents at the 40–50 cm profile. Generally, BC application reduced the NO_3_^−^-N contents of the profile, and this effect was larger under N420 + BC(W) treatment.

Total N contents gradually decreased at the 0–5 cm and 5–10 cm of the soil profile, followed by an increasing trend at 10–30 cm and finally a gradual decrease (Figure 2c). Compared to N420, total N contents significantly increased by 18.5–24.0% and 48.1–48.5% at the 0–5 cm and 40–50 cm profiles, respectively, in BC-application treatments, but decreased at 10–20 cm and 20–30 cm of the profiles. The soil’s total N increased in the deeper soils below 30 cm after BC application. In particular, the N420 + BC(W) treatment had 5.5% and 12.1% significantly higher total N contents than the N420 + BC(R) treatment in the 0–5 cm and 5–10 cm soil profile, respectively. For the 10–30 cm soil profile, two BC-amended treatments subsequently decreased total N contents compared with the N420 treatment. However, the total N content of the 30–40 cm profile under N420 + BC(R) was 23.0% higher than that under N420 + BC(W). Finally, there was no significant difference in TN content between these two treatments in the 40–50 cm profile.

#### 2.1.3. Available P and K Contents

Soil-available-P contents did not change with a similar trend along with the 0–50 cm profile depth (Figure 3a). Compared to N420, application of BC significantly increased the available P contents in all profiles approximately 2-fold. In particular, at the 5–10 cm profiles, N420 + BC(R) and N420 + BC(W) had significantly more available P than N420, respectively. Even below 30 cm of the profile, soil-available-P contents increased by 3.3–10.1% under N420 + BC(W) compared with N420 + BC(R) and N420.

Soil available-K contents were the highest (from 120 to 145 mg kg^−1^) in the 0–5 cm profile, followed by an overall increasing trend (Figure 3b). Compared to N420, there was no significant difference in available-K content at the 0–5 cm and 5–10 cm profiles under the two BC-applied treatments. However, the available-K contents at the 10–20 and 20–30 cm soil profiles under the two BC-applied treatments were 25.9–38.1% and 31.2–57.5% significantly higher than that under N420. Moreover, N420 + BC(W) had a significantly higher available-P content than N420 + BC(R). At the 30–40 cm profile, BC had no significant effect on soil-available-K content; however, up to the 40–50 cm profile, BC applied at rice season significantly increased the available-K contents by 56.1%, compared N420 treatment.

#### 2.1.4. SOM

Overall, we observed a first increasing and then decreasing trend with profile depth for SOM content (Figure 4). BC application significantly increased SOM contents at the 0–5, 5–10, and 40–50 cm profiles compared to the N420 treatment. SOM content reached its maximum at the 0–5 cm profile, which was significantly higher than other profiles. Meanwhile, SOM contents at 0–5 cm of the two BC-added treatments were 57.2–63.1% significantly higher than the N420 treatment. However, BC that was applied at both crop seasons lowered the SOM contents at the 10–20 and 20–30 cm profiles. The SOM in the 40–50 cm profile was significantly increased by 52.8–53.5% with BC application.

### 2.2. Wheat Yield and the Related Agronomic Traits

Application of N fertilizer alone or combined with BC significantly increased wheat yield compared with the control (Table 1). Two BC-added treatments produced same wheat grain as N420. All wheat yield constitutive factors were not changed by BC application in this study. Compared to N420, wheat yield increased by 5.8% and 5.9%, respectively, in the N420 + BC(R) and N420 + BC(W) treatments, though the difference was not statistically significant.

### 2.3. Wheat NUE

From the data in Table 2, we found that N and BC significantly changed the straw and grain N contents. Meanwhile, BC applied at rice season increased the N uptake in straw by 38.9%, but BC applied at wheat season decreased it by 22.2%. N contents were found to be significantly higher in grain than straw (Table 2). Therefore, there was no significant difference in grain or total N uptake by rice-shoot biomass. There was also no significant difference in the NUE of rice with N application, which changed from 22.7% to 24.9%.

### 2.4. Amino Acid in Wheat Grain

Total amino acids (TAA) in wheat kernels were not significantly affected by BC application (Table 3). The TAA content in each treatment was N420 > N420 + BC(W) > N420 + BC(R). The highest content of essential amino acids (EAA) was leucine, and the highest content of non-essential amino acids (NEAA) was glutamic acid, with the highest values both being seen with N420 treatment. The difference in total EAA among treatments was not significant, except for isoleucine and phenylalanine. Also, total NEAA did not differ significantly among treatments, except for proline. The EAA/TAA ratios in the grains ranged from 35.6% to 35.8% and were not significantly different among treatments. Therefore, BC applied in either wheat or rice season had no significant effect on the amino acid contents of the grains.

## 3. Discussions

### 3.1. Effects of BC Addition on pH and Nutrients in the Soil Profiles

Most studies have shown that the application of BC increased soil pH [27,28,29] because the high-temperature pyrolysis process converts biomass acids into bio-oil fractions, while solid BC inherits its alkalinity, which increases soil pH (also known as the liming effect) [7]. However, studies have that found that soil pH decreased after BC application; therefore, the regulation of soil pH by BC is multidirectional [30]. In this study, it was shown that soil profile pH decreased after BC application. It is possible that the surface ash degradation during the natural aging process after BC is input into the soil, especially oxidation and acidification, leads to an increase in surface acidic oxygen-containing functional groups such as carboxyl (–COOH) and hydroxyl (–OH) groups [31]. Although the pH change in the surface soil was not extensive, the pH change in the deeper soil was significant, indicating that BC has an obvious migration process to the deeper soil [24], which lead to a decrease in pH in the deeper soil. In the 0–40 cm profile, the effects of BC application on soil pH were basically the same as in the rice and wheat seasons; however, in the 40–50 cm profile, the application of BC lowered soil pH more when being applied in the rice season than the wheat season, probably because the moisture of the flooding management measures in the rice season promoted the downward migration and accumulation of BC particles in the soil [26]. Nevertheless, soil pH changes are regulated by a number of factors, and although the application of BC in this study led to a decrease in pH in the observed soil profiles, it remained in the neutral range and did not stress the growth of crops such as rice, wheat, and maize [28].

The current study showed that the effect of BC application in rice or wheat season on NH_4_^+^-N and NO_3_^−^-N contents in soil profile were observed in the top (0–20 cm) soil layer. For example, compared to N420, N420 + BC(W) significantly reduced NO_3_^−^-N contents in 0–5 cm of soil profile, and the application of BC also reduced the NH_4_^+^-N contents in the 0–20 cm profile, with the N420 + BC(W) having a more significant reduction effect. The soil profile NH_4_^+^-N contents in this experiment were lower than the NO_3_^−^-N contents, suggesting that the BC application changed soil conditions and rendered mineralizable NH_4_^+^-N virtually nonexistent in the soil, as it was all converted to NO_3_^−^-N [32]. In addition, the application of BC significantly reduced the NH_4_^+^-N and NO_3_^−^-N contents in the soil profiles, suggesting that BC produced at high temperatures (>600 °C) has the potential to adsorb NH_4_^+^ and NO_3_^−^ ions because of its large specific surface area and the presence of net negative and positive charges [33]. Similarly, a previous study has shown that most BC has a negatively charged surface and, therefore, has a strong retention capacity for NH_4_^+^ [34]. Overall, BC application significantly affected NH_4_^+^-N, NO_3_^−^-N, and total N contents in all the investigated soil profiles. Compared to N420, BC application significantly increased total N contents in the 0–10 and 40–50 cm profiles, probably because the addition of BC raised SOC and consequently resulted in a higher C/N. Since the C/N ratio is so large, microbial decomposition and mineralization is slow, and so it consumes more available N (NH_4_^+^-N and NO_3_^−^-N) in the soil and promotes the capacity of the soil to absorb and hold N and other nutrient elements. It was shown that BC not only has a strong adsorption capacity, but also can increase soil total N nutrients and effectively reduce N losses, thus delaying nitrogen release. Lehmann et al. [35] found that the application of BC in the growing period of the crop significantly reduced the leaching of NH_4_^+^-N from the soil. The present study confirmed this effect of BC application. Furthermore, BC-amended treatments had higher total N contents at the 40–50 cm soil profile, suggesting that BC moved to the deeper soils to sequester nutrients or that BC migrated in response to environmental factors including wind, runoff, infiltration, and soil tillage managements [36,37,38]. Furthermore, the unique wet and dry cycle of water management in rice–wheat rotation caused dramatic changes in soil physicochemical and BC properties, which affected the conversion of soil N during the two cropping seasons [39]. Our results showed that N420 + BC(W) treatment significantly decreased NH_4_^+^-N contents the 5–10 and 10–20 cm, and NO_3_^−^-N contents in 0–5 and 20–30 cm profiles but increased the total N contents in the 0–5 and 40–50 cm profiles (Figure 2). Therefore, choosing to apply BC in the wheat season is beneficial to absorbing NH_4_^+^-N and NO_3_^−^-N and increasing total soil N nutrient to achieve full utilization of N fertilizer and lower N leaching losses.

It is well known that BC can adsorb, load, release, and increase the slow-release effect of soil nutrients [40] as well as can retain nutrients and prolong their residence time in the soil to achieve the effect of maintaining soil fertility [41]. There were differences in the effects of BC on soil nutrients, with some studies showing that N fertilizer and BC combination increased the available soil P and K contents [40,41], which was also confirmed by the present study. What is more, Liu et al. [42] found that BC altered the distribution of available P in 0–20 cm soil by increasing the adsorbing capacity of P in soil. In the present study, BC application increased the available P in all soil profiles by nearly 2-fold of that in the N420 treatment. The effect of BC application on soil-available-P contents was also seasonally related and reflected by the N420 + BC(W) treatment significantly increasing the available-K content in the 10–20 cm and 20–30 cm profiles compared to the N420 treatment.

As a key component of soil fertility in agro-ecosystems, increases in SOM content improve soil aggregation and promote soil nutrient effectiveness [43]. In particular, we found that N420 + BC (W) increased the SOM content in the 0–5 cm and 40–50 cm profiles by 63.1% and 52.8%, respectively (Figure 4). All the aforementioned data indicate that BC application could overall increase the available nutrients across the 0–50 soil profile, thus enhancing soil fertility. In addition, the effects of the BC applied in the wheat season to increase the nutrients in the soil profile were more pronounced than those seen when it was applied in the rice season. Previous research has demonstrated that BC could decrease the negative impacts on the soil under alternating wet and dry conditions and increase the retention of nutrients as well as the soil moisture [44,45]. Therefore, BC application can increase soil profile nutrients, especially the available nutrients, and is more suitable for application under drought conditions (i.e., in the wheat season) to reach the optimal efficacy of BC.

### 3.2. Effects of BC Addition on Wheat Yield, N Uptake, NUE, and Amino Acids

The essence of the crop yield formation is the accumulation and distribution of dry matter, the amount of N fertilizer, and the input of exogenous organic materials, which affect the accumulation of dry matter in the aboveground part of crops. Therefore, reasonable N fertilizer and BC distribution can help to improve dry matter formation and N transport in crops to help produce a high yield of wheat [46]. The present study showed that compared to the control, the N and BC combination was able to promote an increase in wheat yield and its constitutive factors. Compared to N420, BC application had an increasing effect on wheat yield and although it did not have a significant effect, it is still of great importance for food security, particularly in China [45].

Both N content and its uptake rate in grain under each treatment were greater than that in straw, indicating that more N was absorbed and utilized by wheat grain. BC was able to enhance the soil’s ability to supply N to meet the crop’s N demand in the key period of yield formation (grain filling period) and, at the same time, promote the transfer of N in the stalks to the grains [47]. Wheat grain harvested from N420 + BC(W) had significantly higher N contents than that from in N420 + BC(R). However, there was no significant difference in grain N uptake and NUE. Crop NUE not only reflects the uptake and utilization of N fertilizer, but is also an important indicator for evaluating N-fertilizer management [48]. There are many factors affecting NUE, which are subject to climatic factors, soil conditions, and crop varieties in addition to fertilizer. Meanwhile, the results of short-term experiments are also very different from those of long-term experiments. Generally, BC has a positive effect on enhancing crop yield NUE [49]. Although BC and N treatments did not have a significant effect on NUE in the present study, NUE was found to be slightly higher in the N420 + BC(R) treatment than in the N420 + BC(W) treatment. This indicates that there is a cumulative effect of BC application, which increases NUE in the rice season; meanwhile, application of BC in the wheat season may exceed the crop utilization limit, resulting in a weakening of its effect [50]. In addition, this study was a short-term experiment, and NUE was again affected by a variety of factors which could not be effectively illustrated in the final effect of BC on NUE. Finally, it is more reasonable to apply BC during the wheat season, taking into account the fertility characteristics of the soil profile and the uptake of N by wheat.

The degree of balance and level of essential amino acid composition in wheat grain determines the nutritional quality of protein and is directly related to the degree of utilization of nutrients in wheat [51]. The nutritional value of proteins is evaluated mainly on the basis of the contents of essential amino acids. Threonine and lysine are limiting nonessential amino acids that determine a good quality of protein. The application of BC in this study had almost no effect on the content of each amino acid, and the content of limiting amino acids was low, which indicated that BC had no effect on the quality of wheat grain proteins. Furthermore, the EAA/TAA values in this study were in the range of 34–37% and the EAA/NEAA values were in the range of 51–58%, which is close to the required ratios of amino acids (EAA/TAA 40% and EAA/NEAA 60%) recommended by the FAO/WHO for daily intake in humans. The application of BC in either rice or wheat seasons did not affect the wheat grain quality, but it showed that the EAA was slightly larger in N420 + BC(W) than N420 + BC(R). Therefore, BC applied in wheat season can potentially help to ensure the quality of wheat grain (amino acid content).

## 4. Materials and Methods

### 4.1. Experiment Setup

#### 4.1.1. Background Information and Soil Column Setup

The soil column simulation experiment was carried out in the research greenhouse of Jiangsu Academy of Agricultural Sciences (32°08′ N, 118°82′ E). The greenhouse had a roof, and the ventilation conditions were close to those in the field. The study area belongs to the East Asian monsoon climate zone, with an annual precipitation of 1106.5 mm, and an average annual temperature of 15.5 °C. The test soil was taken in May 2018 in the order of 0–20, 20–40, and 40–60 cm profiles from a typical paddy field, which is located in Zhoutie Town, Yixing City, Jiangsu Province, China, with an area of about 15 ha. The soil was homogenized and mixed according to the profile, then dried naturally for about 10 days, sieved through a 2 mm sieve, and filled into soil columns at a capacity weight similar to the site conditions. The soil columns were made of PVC material with a height of 60 cm and a diameter of 30 cm. The soil at different profile levels was filled back into the soil columns according to the sampling orders, tamped, and watered with deionized water to settle the soil [52]. The test soil type was paddy soil, and the physicochemical properties of the 0–20 cm topsoil were: pH (1:2.5 soil–water ratio) 6.36; available N 0.38 g kg^−1^; total N, P, and K contents 1.56 g kg^−1^, 0.96 g kg^−1^ and 4.12 g kg^−1^, respectively; SOM 22.8 g kg^−1^; C/N ratio 8.48; and cation exchange capacity (CEC) 19.6 cmol kg^−1^. The wheat straw was cracked at a high temperature of 500 °C to make the biochar, which had a pH of 9.80, total N of 8.1 g kg^−1^, total carbon of 67.5%, and a BET surface area 32.0 m^2^ g^−1^.

#### 4.1.2. Experimental Design and Management

A total of four treatments (three replicates per treatment) were set up: no N fertilizer and BC addition (control); N fertilizer (420 kg ha^−1^ yr^−1^) only (N420); same N fertilizer (420 kg ha^−1^ yr^−1^); and added BC (5 t ha^−1^) applied to the rice [N420 + BC(R)] or wheat [N420 + BC(W)] seasons. Meanwhile, a control treatment was observed for calculating the wheat NUE. Nitrogen fertilizer application in this experiment was 420 kg ha^−1^ yr^−1^ in the rice–wheat rotation system, of which 240 kg ha^−1^ yr^−1^ was in the rice season and 180 kg ha^−1^ yr^−1^ was in the wheat season. BC was applied at a rate of 5 t ha^−1^ (equivalent to 35 g pot^−1^) into the 0–15 topsoil before transplanting seedlings in the rice season (R) and before sowing in the wheat season (W), respectively. The experiment was started in June 2019 and ended in May 2022. The effects of BC applied in the different seasons for three consecutive years on wheat yield, quality, and post-harvest soil profile physicochemical properties were investigated after wheat harvest in the last wheat season. The experiment was conducted in a conventional water and fertilizer management mode, with N fertilizer supplied by urea (46% N content) in the ratio of 30%:30%:40% as basal, tiller, and spike fertilizers during both the rice and wheat seasons. Both P and K fertilizers were applied as a single application as basal fertilizers in the forms of calcium superphosphate and potassium chloride at rates of 90 kg P_2_O_5_ ha^−1^ and 120 kg K_2_O ha^−1^. The tested rice (*Oryza sativa* L., var. Nangeng 46) was transplanted in June 2021, with three holes per soil column and three seedlings per hole, and was harvested in November. For the tested wheat (*Triticum aestivum* L. var. Ningmai 26), 50 wheat seeds were evenly sown into each 5 cm-deep soil column in November 2021 and grew from December 2021 to May 2022. Agronomic management practices, including irrigation as well as weed and pest control, were carried out during the trial according to the traditional practices of local farmers.

### 4.2. Sample Collection and Measurement

#### 4.2.1. Soil Sampling and Analysis

Soil samples were collected in May 2022 after wheat harvest by soil auger in each soil column in layers (0–5, 5–10, 10–20, 20–30, 30–40, and 40–50 cm) according to a five-point sampling method, with soil from the same layers mixed into one sample. After removing debris and any visible roots, each sample was sieved (<2 mm) and homogenized for further analysis.

A total of 10 g of air-dried soil samples (<2 mm) was taken and left to stand for 30 min with 25 mL of water, and then the soil pH was determined using a pH meter. A total of 5 g of fresh soil sample was added to 25 mL of 2 mol L^−1^ KCl solution, shaken for 1 h, and then filtered. NH_4_^+^-N and NO_3_^−^-N were determined by indophenol blue colorimetry and UV spectrophotometry; 1.000 g of air-dried soil sample (<0.15 mm) was taken, 2 g of accelerator was added and moistened with purified water, 5 mL of H_2_SO_4_ was added and put into a digestion tube to digest, and then the soil’s total N was determined by Kjeldahl N determination after digestion. The available AK was determined by 1 mol L^−1^ NH_4_OAc using the leaching-flame photometric method, and the available P was determined by 0.5 mol L^−1^ NaHCO_3_ using the leaching-molybdenum antimony colorimetric method. Soil organic matter was determined by K_2_Cr_2_O_7_ using the oxidation-external heating method. The above methods were adopted from the conventional methods of Soil Agrochemical Analysis.

#### 4.2.2. Wheat Yield and NUE

Straw and grain were harvested separately at the maturity stage of wheat; spike number and kernels per spike in each soil column were recorded; meanwhile, straw and seed weights were determined. Taken back to the laboratory, twelve healthy plants were selected for each treatment and their plant height, straw and seed weight, and number of grains were recorded. The remaining plants were threshed and their straw and seed weights were recorded. The straw and seeds of the selected plants were put into an oven at 105 °C for 0.5 h to kill the green, and then dried at 75 °C to a constant weight. The straw and grain of the remaining plants were put into the sunlight to dry naturally. The dried straw and seeds were crushed through a 0.25 mm sieve, and the N content of straw and seeds was determined by the Kjeldahl method. Harvest index, straw and grain N uptake, and NUE were calculated according to the following equations.
Harvest index (%) = dried grain weight (g)/[dried grain weight (g) + dried straw weight (g)] × 100%(1)
Straw/grain N uptake(g) = dry weight of straw/grain (g) × N content of straw/grain (g kg^−1^)/1000(2)
N use efficiency (NUE, %) = [N uptake in N application treatment (g) − N uptake in control (g)]/N application rate (g) × 100%(3)

#### 4.2.3. Amino Acid

The amino acid content of wheat grain was determined using a fully automatic amino acid analyzer, and amino acids and their component contents were determined by the GB/T18246-2000 [53] method. EAAs are a sum of threonine, valine, methionine, isoleucine, leucine, phenylalanine, and lysine; additionally, NEAAs are a sum of histidine, aspartate, serine, glutamate, glycine, alanine, cystine, leucine, arginine, and proline. TAAs are the sum of EAAs and NEAAs.

### 4.3. Statistical Analysis

The data were collated and analyzed by ANOVA using Excel 2010 and SPSS 26.0 software (SPSS Inc., Chicago, IL, USA). Multiple comparison tests were performed between the different treatments using Duncan’s method, with different lowercase letters indicating significant differences between treatments at the significance level of *p* < 0.05.

## 5. Conclusions

The addition of BC significantly reduced the NO_3_^−^-N and NH_4_^+^-N contents in the soil profile, and the reduction was greater when BC was applied in the wheat season. Meanwhile, BC applied in the wheat season significantly increased the total N available at 0–10 cm, K at available 20–30 cm, P available at 30–50 cm, and SOM available at 0–5 cm profiles. Moreover, BC application in wheat season could ensure wheat yield and quality as well as promote N uptake in wheat. Therefore, BC application in wheat season is more helpful to improving the nutrient content in the soil profile while also ensuring wheat yield and quality. Nevertheless, this experiment is a short-term soil column simulation experiment and, as such, the long-term effects of BC application in different seasons on soil profile nutrient distribution, crop yield, and crop quality need to be further investigated, especially at the field scale.

## Figures and Tables

**Figure 1 plants-12-04131-f001:**
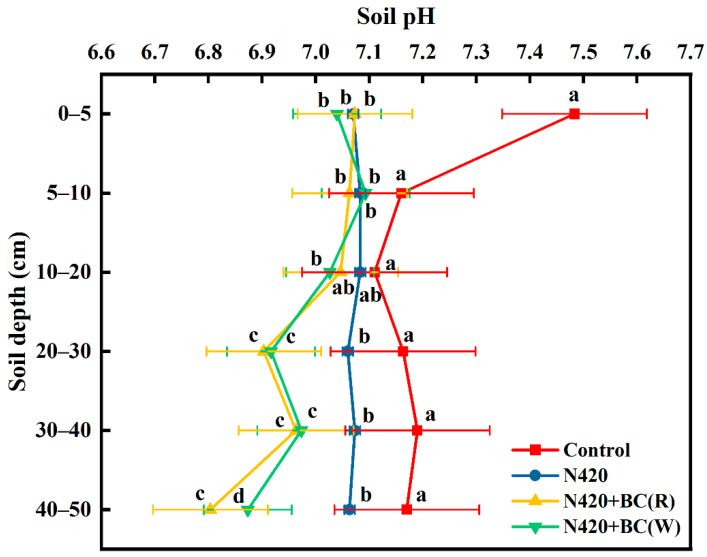
Effects of biochar (BC) application in rice (R) or wheat (W) seasons on the pH at soil profile. The values are means ± SD (*n* = 3). Error bars represent the SD of the mean of three replicates. Different letters mean statistically significant differences at *p* < 0.05.

**Figure 2 plants-12-04131-f002:**
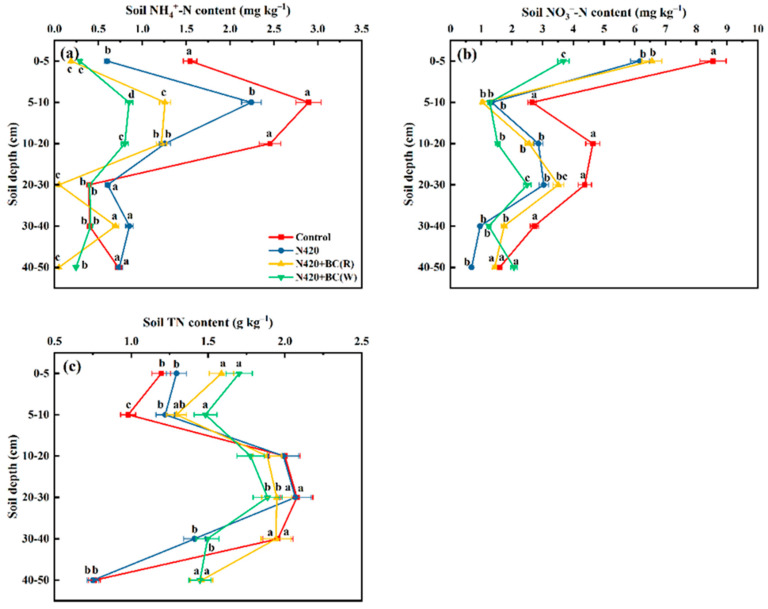
Effects of biochar (BC) application in rice (R) or wheat (W) seasons on contents of ammonium (NH_4_^+^-N, **a**), nitrate (NO_3_^−^-N, **b**), and total N (TN, **c**) in soil profile. The values are means ± SD (*n* = 3). Error bars represent the SD of the mean of three replicates. Different letters mean statistically significant differences at *p* < 0.05.

**Figure 3 plants-12-04131-f003:**
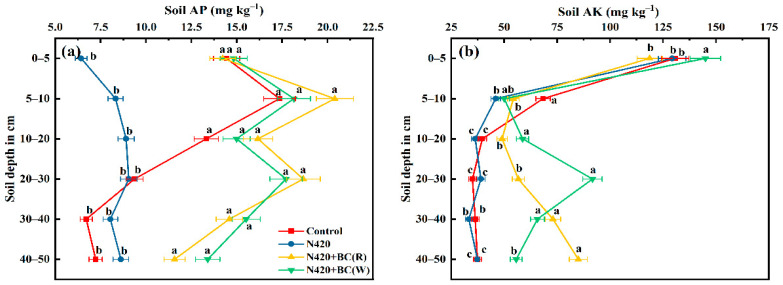
Effects of biochar (BC) application during rice (R) or wheat (W) seasons on available phosphorus (AP, **a**) and available potassium (AK, **b**) in soil profiles. The values are means ± SD (*n* = 3). Error bars represent the SD of the mean of three replicates. Different letters mean statistically significant differences at *p* < 0.05.

**Figure 4 plants-12-04131-f004:**
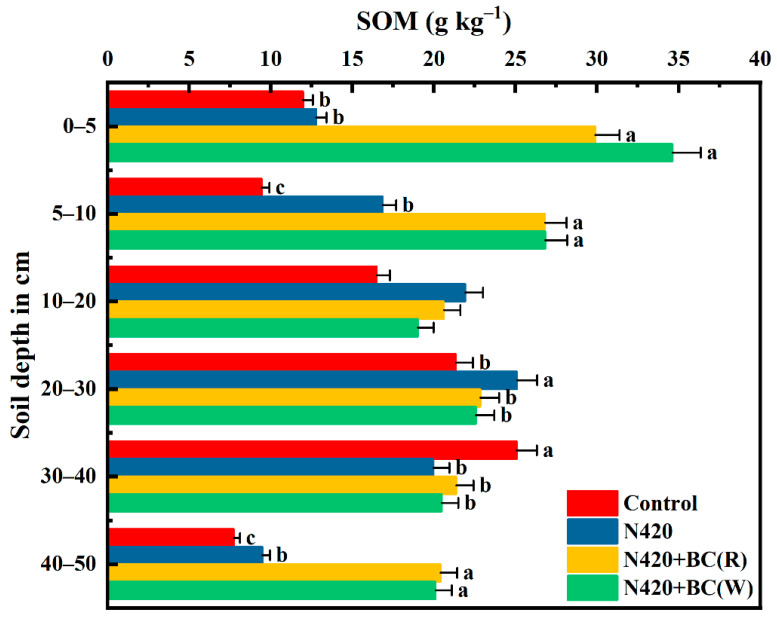
Effects of biochar (BC) application during rice (R) and wheat (W) seasons on soil organic matter (SOM) in the soil profiles. The values are means ± SD (*n* = 3). Error bars represent the SD of the mean of three replicates. Different letters mean statistically significant differences at *p* < 0.05.

**Table 1 plants-12-04131-t001:** Effects of biochar (BC) application during rice (R) and wheat (W) seasons on wheat yield and its components in 2022.

Treatment	Straw Biomass (g pot^−1^)	Grain Yield (g pot^−1^)	Spike Number	Kernels Per Spike	Thousand-Kernel Weight (g)	Theoretical Wheat Yield (g pot^−1^)	Harvest Index (%)
Control	19.7 ± 2.6 b	10.2 ± 1.0 b	33.3 ± 2.5 a	10.3 ± 1.0 b	41.1 ± 4.5 b	14.3 ± 3.7 b	34.3 ± 1.2 b
N420	58.5 ± 8.5 a	44.5 ± 5.4 a	36.3 ± 2.5 a	28.7 ± 1.5 a	50.4 ± 4.1 a	52.4 ± 2.2 a	43.3 ± 1.7 a
N420 + BC(R)	60.1 ± 2.7 a	43.3 ± 3.2 a	35.3 ± 0.6 a	31.2 ± 4.6 a	50.4 ± 1.7 a	55.7 ± 9.2 a	41.9 ± 0.9 a
N420 + BC(W)	56.2 ± 1.0 a	40.4 ± 2.3 a	37.0 ± 2.7 a	28.7 ± 1.0 a	52.5 ± 0.5 a	55.7 ± 2.8 a	41.8 ± 1.5 a

Note: Values are means ± SD (*n* = 3) and different letters in the same column indicate statistically significant differences at the level of *p* < 0.05.

**Table 2 plants-12-04131-t002:** Effects of biochar (BC) application during rice ^®^ and wheat (W) seasons on nitrogen (N) content, uptake, and N use efficiency (NUE) of wheat straw and grain.

Treatment	N Content (g kg^−1^)		N Uptake (g pot^−1^)			NUE (%)
	Straw	Grain	Straw	Grain	Total	
Control	2.6 ± 0.1 c	22.0 ± 0.3 a	0.05 ± 0.01 d	0.22 ± 0.02 b	0.28 ± 0.02 b	–
N420	3.1 ± 0.1 b	18.5 ± 1.4 b	0.18 ± 0.03 b	0.82 ± 0.04 a	1.00 ± 0.07 a	23.2 ± 1.6 a
N420 + BC(R)	4.2 ± 0.1 a	18.8 ± 0.7 b	0.25 ± 0.01 a	0.82 ± 0.09 a	1.07 ± 0.10 a	24.9 ± 2.4 a
N420 + BC(W)	2.5 ± 0.2 c	20.8 ± 2.7 ab	0.14 ± 0.02 c	0.84 ± 0.09 a	0.98 ± 0.07 a	22.7 ± 1.7 a

Note: Values are means ± SD (*n* = 3) and different letters in the same column indicate statistically significant differences at *p* < 0.05.

**Table 3 plants-12-04131-t003:** Effects of biochar (BC) application during rice (R) and wheat (W) seasons on amino acid content in wheat grains.

	Amino Acid Components	Amino Acid Content (%)		
		N420	N420 + BC(R)	N420 + BC(W)
EAA	Threonine	0.23 ± 0.03	0.25 ± 0.02	0.23 ± 0.03
	Valine	0.41 ± 0.04	0.45 ± 0.03	0.42 ± 0.05
	Methionine	0.15 ± 0.01	0.16 ± 0.01	0.16 ± 0.00
	Isoleucine	0.30 ± 0.02 a	0.23 ± 0.01 b	0.25 ± 0.02 b
	Leucine	0.52 ± 0.08	0.50 ± 0.01	0.49 ± 0.01
	Phenylalanine	0.34 ± 0.02 b	0.44 ± 0.04 a	0.36 ± 0.02 b
	Lysine	0.28 ± 0.04	0.24 ± 0.01	0.24 ± 0.01
ΣEAA		2.25 ± 0.10	2.19 ± 0.03	2.21 ± 0.13
NEAA	Aspartate	0.59 ± 0.04	0.57 ± 0.03	0.56 ± 0.03
	Serine	0.35 ± 0.03	0.37 ± 0.03	0.38 ± 0.02
	Glutamate	1.43 ± 0.09	1.32 ± 0.02	1.36 ± 0.02
	Proline	0.19 ± 0.00 a	0.17 ± 0.00 b	0.18 ± 0.01 ab
	Glycine	0.24 ± 0.01	0.26 ± 0.03	0.24 ± 0.02
	Alanine	0.39 ± 0.01	0.39 ± 0.01	0.39 ± 0.01
	Cystine	0.10 ± 0.00	0.12 ± 0.01	0.11 ± 0.01
	Leucine	0.12 ± 0.01	0.12 ± 0.01	0.12 ± 0.00
	Histidine	0.11 ± 0.01	0.14 ± 0.03	0.11 ± 0.00
	Arginine	0.48 ± 0.03	0.51 ± 0.01	0.49 ± 0.03
ΣNEAA		4.04 ± 0.02	3.93 ± 0.07	3.99 ± 0.13
ΣTAA		6.29 ± 0.10	6.12 ± 0.06	6.19 ± 0.19
EAA/TAA		35.7 ± 1.00	35.8 ± 0.60	35.6 ± 1.50

Note: The values are means ± SD (*n* = 3). Different letters mean statistically significant differences at *p* < 0.05.

## Data Availability

Data are available after requesting.

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
