# Peer review of "Effects of Biochar Applied in Either Rice or Wheat Seasons on the Production and Quality of Wheat and Nutrient Status in Paddy Profiles"

_plants, 2023, doi:10.3390/plants12244131_

Round 1

Reviewer 1 Report

Comments and Suggestions for Authors

This publication presents interesting results related to the effects of biochar supplementation during rice or wheat seasons on the soil profile nutrient status and the production and quality of wheat. The adopted approach could provide more insights into the use of biochar to improve soil and plant yield quality. different

The manuscript was well introduced; however the methodology and results description should be rechecked. Thereby, the manuscript needs extensive revisions to be suitable for publication in Plants.

General comment

Comment 1: The lines numbering is missing; it was difficult to provide the comments without!

Comment 2: The English of this manuscript needs extensive revision.

Comment 3: Key points in methodology are not provided or are misleading.

Comment 4: Results description should be revised.

Other comments

- Title: please change the title to “Effect of biochar applied at either rice or wheat seasons on the production and quality of wheat, and nutrient status in paddy soil profiles”.

Abstract

- Please shorten the abstract.

- Keywords: please add “wheat”,

Introduction

- Please change “especially with the prolongation” to “especially when the prolongation”.

- Please add a clear hypothesis at the end of the introduction section.

Results

- Please describe the starting point values before any variation since they are mostly different between the applied treatments.

- Please provide the significance of the treatment abbreviations and other abbreviation included in the figures in the caption.

- Please provide the significance of abbreviations included in tables at the footnote.

- Why did you change the graph type in figure 4 to histogram?

- Figure 4 caption: please change “matter (SOM) in profile” to “matter (SOM) in soil profiles”.

- Please check all results description. Per example, some statements in Table 2 description is not correct.

- Please provide the significance of the abbreviations at the first appearance in the text and then use only abbreviations.

- Please add another column before “Amino acid component” in table 3 to discriminate EAA from NEAA.

Discussion

- Please correct “However, have studies found that soil pH”.

- “The present study was in agreement with it and the application of BC at 40–50 cm profile significantly increased soil TN contents”, please correct this statement since you did not applied BC at 40–50 cm.

M&M

- Please provide C/N for the top soil layer.

- “A total of four treatments (three replicates per treatment) were set up: no nitrogen (N) fertilizer and BC addition (control), N fertilizer (N420) and BC (5 t ha–1) applied at rice [N420+BC (R)] or wheat [N420+BC (W)] seasons at the same rate of N fertilizer (420 kg ha–1 yr–1)“, please reformulate it is not clear.

- “[N420+BC (R)] or wheat [N420+BC (W)] seasons at the same rate of N fertilizer (420 kg ha–1 yr–1); which control was for calculating the nitrogen use efficiency (NUE) of wheat in each treatment. Nitrogen application in this experiment was 420 kg ha–1year–1 in the rice–wheat rotation system, of which 240 kg ha–1year–1 in the rice season and 180 kg ha–1year–1 in the wheat season.”, I do not understand this part, you have [N420+BC (R)] and [N420+BC (W)] and you stated that “Nitrogen application in this experiment was 420 kg ha–1year–1 in the rice–wheat rotation system, of which 240 kg ha–1year–1 in the rice season and 180 kg ha–1year–1 in the wheat season”.

- Did you transplant rice into the pots (PVC tubes)?

- How did you apply BC? Did mix it with soil from 0 to 15 cm?

- Nothing is mentioned about Plant seed treatment and preparation? Did put seeds or seedlings in the pots?

- In subsection 4.2.1, you state “removing stones, debris,”, how can you have stone since the soil is already sieved?

- Please check “4.3 Statistical analysis” subsection.

Comments on the Quality of English Language

The English language of this manuscript needs moderate revision.

Author Response

See attached Response Letter. Thanks.

Reviewer 2 Report

Comments and Suggestions for Authors

The submitted manuscript to PLANTS-MDPI entitled “Effect of biochar applied at either rice or wheat seasons on the production and quality of wheat, and nutrient contents in paddy profiles” is interesting to investigate. BUT, following are the comments that need to be addressed:

There should be line numbering to follow!!!

Abstract Line 3: in soils?

Why did the authors not mention the application of Nitrogen in the title even though they mainly focused on N and BC?

Figure 4: The direction is not appropriate.

It would be very nice if authors could provide some data of relevant genes.

Please rewrite the first sentence of conclusion.

Author Response

Please see attached Response Letter. Thanks.

Round 2

Reviewer 1 Report

Comments and Suggestions for Authors

The authors satisfied all raised comments. I endorse the publication of the current cversion of the manuscript.

Reviewer 2 Report

Comments and Suggestions for Authors

It can be accepted!